# Pediatric Neuroendocrine Neoplasms: Rare Malignancies with Incredible Variability

**DOI:** 10.3390/cancers14205049

**Published:** 2022-10-15

**Authors:** Jennifer T. Castle, Brittany E. Levy, Aman Chauhan

**Affiliations:** 1Department of Surgery, Markey Cancer Center, University of Kentucky, 800 Rose Street, Lexington, KY 40536, USA; 2Department of Surgery, University of Kentucky, 800 Rose Street, Lexington, KY 40536, USA; 3Department of Internal Medicine-Medical Oncology, Markey Cancer Center, University of Kentucky, 800 Rose Street, Lexington, KY 40536, USA

**Keywords:** pediatric, neuroendocrine neoplasms, neuroendocrine tumors, neuroendocrine carcinomas

## Abstract

**Simple Summary:**

Neuroendocrine neoplasms are increasing in incidence at a remarkable rate meaning more providers are encountering them in both adult and pediatric patients. This classification of neoplasm encompasses a wide range of different malignancies with a variety of symptoms at presentation and each treated differently. Additionally, over the past few years there has been a change in classification of these neoplasms and a variety of changes and advances in how they are treated. Given this and their rarity in pediatric patients, healthcare providers may not be familiar with these changes. Our goal with this review was to provide an overview of all the most commonly encountered forms of neuroendocrine neoplasms in pediatric patients with up to date recommendations so any healthcare provider can quickly and accurately acclimate themselves.

**Abstract:**

Neuroendocrine neoplasms (NENs) encompass a variety of neuroendocrine tumors (NETs) and neuroendocrine carcinomas (NECs) which can arise anywhere in the body. While relatively rare in the pediatric population, the incidence of NENs has increased in the past few decades. These neoplasms can be devastating if not diagnosed and treated early, however, symptoms are variable and can be indolent for many years. There is a reported median of 10 years from the appearance of the first symptoms to time of diagnosis. Considering some of these neoplasms have a mortality rate as high as 90%, it is crucial healthcare providers are aware of NENs and remain vigilant. With better provider education and easily accessible resources for information about these neoplasms, awareness can be improved leading to earlier disease recognition and diagnosis. This manuscript aims to provide an overview of both the most common NENs as well as the rarer NENs with high lethality in the pediatric population. This review provides up to date evidence and recommendations, encompassing recent changes in classification and advances in treatment modalities, including recently completed and ongoing clinical trials.

## 1. Introduction

Neuroendocrine neoplasms (NENs) originate from neuroendocrine cells which can be found throughout the body. As such, NENs can develop anywhere neuroendocrine cells are in the body, but are most commonly found in the lungs, pancreas, and gastrointestinal tract [1]. The classification of NENs is inconsistent amongst organ system of origination. The World Health Organization (WHO) has provided some clarity and more consistent guidelines for the grading of NENs in their most updated reports and has been reflected in the recent National Comprehensive Cancer Network (NCCN) guideline updates [2]. However, classification and grading for NENs remain organ specific to a certain degree. Gastroenteropancreatic NENs are divided into two groups based on differentiation: neuroendocrine tumors (NETs) and neuroendocrine carcinomas (NECs). NECs are poorly differentiated tumors with mitotic rate greater than 20 mitoses per 2 mm^2^ and Ki-67 index greater than 20%. NETs are well-differentiated lesions which encompass the remainder of the NENs and are further divided into three grades: grade 1 (no necrosis, mitotic rate less than 2 per 2 mm^2^, Ki-67 less than 3%), grade 2 (necrosis present or mitotic rate of 2–20 mitosis per 2 mm^2^, Ki-67 between 3–20%), and grade 3 (well differentiated with mitotic rate of greater than 20 mitosis per 2 mm^2^ or Ki-67 greater than 20%) [3,4]. Many of the other organ systems follow this same classification system. Contrarily, bronchopulmonary NENs, while still separated based on mitotic rate and differentiation, are named differently and irrespective of Ki-67 proliferation index (typical, atypical, and carcinoma) (Table 1) [3,5,6]. In 2018, the International Agency for Research on Cancer and a WHO expert consensus meeting proposed a uniform classification scheme irrespective of site of origin to create more consistency across organ systems [7]. While their proposed system to dichotomize all NENs into either NET of NEC for every organ system has not been implemented, this may represent the future direction with subsequent classification updates.

Separate but related to classification, the staging systems for some of the organ systems have also changed over the past few years. The 8th edition of the American Joint Committee on Cancer (AJCC) separated pancreatic NEN staging to reflect the difference in tumor biology and prognosis between pancreatic NETs and pancreatic NEC. In this revision, pancreatic NEC still fall under the staging system of other exocrine pancreatic tumors, however, a new staging system was proposed for pancreatic NETs in accordance with their more benign nature [8]. As the scientific community is better able to delineate the clinical course and prognoses of NENs, their classification and staging systems continue to evolve.

Given the complexity and rarity within this disease in the pediatric population, healthcare providers may not be familiar with NENs. Our goal with this review was to provide an overview of commonly encountered forms of NENs in pediatric patients with up to date recommendations so any healthcare provider can quickly and accurately acclimate themselves.

## 2. Epidemiology

NETs are more thoroughly characterized in the adult population with pediatric focused research lacking due to its rarity in this population. Overall, the incidence of NENs has grown by almost 7-fold from 1973 to 2012 [9]. Given the advances and more frequent use of imaging, the incidence of pediatric NENs is suspected to have continued to increase since 2011, similar to the adult population, but an updated pediatric-specific epidemiological study is lacking in the literature. Based on the National Cancer Institute’s Surveillance, Epidemiology, and End Results (SEER) data, pediatric NENs are most-commonly found in the gastrointestinal tract (i.e., appendix), lung (one of the most common primary lung neoplasms in children) and breast, but have been reported in a variety of locations throughout the body (Figure 1) [10,11,12]. However, the appendix has historically been thought of as the overwhelmingly most common NEN in children, which has been demonstrated to account for 80% of all pediatric NENs in other published studies [13].

## 3. Diagnosis

### 3.1. Presentation

NENs are notoriously indolent with vague initial symptoms that present a median of almost ten years prior to diagnosis [14]. This is ubiquitous in both the adult and pediatric population leading to delays in diagnosis in both. It is important to recognize that patients diagnosed as young adults may have had symptoms for many years, in which case they may have developed symptoms in adolescence and childhood. Given their proclivity to go unnoticed for years, a substantial proportion of patients have metastatic disease at presentation. Overall in children, 22% have regional spread, 10% have distant disease, and 5% have an unknown primary site at time of diagnosis [12]. Similar to the spectrum of diseases that are encompassed in pediatric NENs, metastatic rate ranges greatly depending on the location of the primary lesion. Appendiceal NENs rarely ever metastasize where as other gastroenteropancreatic NENs are metastatic at time of diagnosis in 50% of children [15].

Symptoms of NENs, when they do manifest, are dependent on the location as well as the functional secretory status of the tumor. Appendiceal NENs in children are typically non-secretory and diagnosed on pathology after resection for presumed appendicitis. These children typically present rather acutely with only days of symptoms consistent with appendicitis (e.g., abdominal pain, fever, and anorexia) [16,17,18]. Additionally, imaging, whether it be ultrasound or CT, is usually consistent with acute appendicitis [19]. Similarly, they can have an associated leukocytosis, elevated c-reactive protein, and/or thrombocytosis. But, around half of the reported cases had no laboratory abnormalities [19]. Along the natural course of treatment for presumed appendicitis, they undergo appendectomy and pathology later reveals appendiceal NEN. Other extra-appendiceal gastrointestinal NENs, can present with nondescript abdominal pain and/or diarrhea [15]. Bronchopulmonary NENs are notorious for presenting as a cough with recurrent pneumonias that then progress to wheezing and difficulty breathing [20,21,22]. These symptoms can go on incorrectly diagnosed as asthma given the similarity in symptoms and age of onset in the pediatric population [11].

When functional, NENs have symptoms secondary to hormone hypersecretion. The classically associated hypersecretion syndrome associated with NENs is carcinoid syndrome. Symptoms of carcinoid syndrome include diarrhea, difficulty breathing, and flushing typically secondary to hypersecretion of serotonin [23]. Previously it was thought that carcinoid syndrome only evolved from hepatic metastasis allowing for direct systemic vascular injection of the hormones from the tumors [23]. However, carcinoid syndrome and one of its more clinically detrimental complications, carcinoid heart disease, has been reported in patients with no evidence of hepatic metastasis [24,25]. Although well documented in the adult population, there are few cases of carcinoid syndrome in the pediatric literature [13,26,27]. In addition to carcinoid syndrome, some NENs are associated with ectopic Cushing’s syndrome, a condition characterized by excess glucocorticoids from hypersecretion of adrenocorticotropic hormone (ACTH). Symptoms include growth deceleration, truncal obesity, facial plethora, high blood pressure, and weight gain [28]. Despite NENs being the most common cause of ectopic ACTH overproduction in the pediatric population, occurrences remain infrequent [29]. Of the pediatric reported cases, most ectopic ACTH-producing NENs are bronchial or pancreatic [22,30]. Pituitary adenomas are the overall most-common source of ACTH overproduction (non-ectopic) in children and have recently been reclassified as neuroendocrine neoplasms by the WHO [28,31,32].

The pancreas is the site of a multitude of different functional NENs. Gastrin production results in Zollinger-Ellison syndrome which can present as severe, recurrent epigastric pain and malabsorption from gastric and duodenal ulcers and diarrhea. NENs in the duodenum and the pancreas have been documented to produce gastrin [15,33]. In children, the gastrin-producing NENs (gastrinoma) and insulin-producing NENs (insulinoma) are the most common functional pancreatic NENs [15]. Similar to adults, insulinomas in children present with hypoglycemic symptoms (i.e., sweating, light-headedness, confusion, palpitations) [34,35]. Other functional pancreatic NENs have been described in the literature but are exceedingly rare. VIPomas, defined by hypersecretion of vasoactive intestinal peptide (VIP), present with severe dehydration and malabsorption from uncontrollable diarrhea, but are much rarer in the pediatric population [36]. Somatostatinomas (hypersecretion of somatostatin) may present in the pancreas or the duodenum but are rare even in the adult population. There has only been sparsely documented occurrences in the pediatric and young adult population [37,38]. Presenting symptoms of somatostatinoma include abdominal pain, gastrointestinal bleeding, and jaundice [37]. With similar infrequency, glucagonomas (pancreatic NEN with hypersecretion of glucagon) have very few documented cases in the literature in pediatric and young adult patients [39,40,41]. Necrolytic migratory erythema, a classic skin rash associated with glucagonomas, is often a presenting feature as well as diabetes and weight loss [42].

Pheochromocytomas and paragangliomas are catecholamine-secreting NENs and responsible for less than 5% of pediatric patients with hypertension [43]. Up to 90% of pediatric patients with a pheochromocytoma or paraganglioma present with hypertension [43]. Some children also experience headaches, palpitations, tremors, and flushing [44]. Pheochromocytomas and paragangliomas originate from the same cell type but differ in that pheochromocytomas are of the adrenal gland and paragangliomas are extra-adrenal tumors. Neuroblastomas, another neuroendocrine tumor frequently originating from the adrenal gland but can also be found anywhere in the sympathetic nervous system, is also associated with catecholamine secretion [45]. However, patients with neuroblastoma typically present with an asymptomatic mass or symptoms secondary to the location of the mass rather than symptoms of excess catecholamines [46]. This can include constipation from bowel compression, hypertension from renal artery compression, and scoliosis with or without neurosensory and motor symptoms from spinal cord compression [47,48,49,50]. Neuroblastomas are heterogenous with courses ranging from benign to very aggressive. Often discussed with neuroblastomas are ganglioneuromas and ganglioneuroblastomas, which are benign and mixed-form tumors, respectively. These tumors are cellularly similar to neuroblastomas but are beyond the scope of this review [51].

Quite distinct from the previously mentioned NENs are Merkel Cell Carcinomas of the skin. This is a very aggressive skin cancer with high mortality rate [52]. There are documented cases in the pediatric population demonstrating variable presentations from simple skin dysplasia to subcuticular masses [53]. There is evidence to suggest a more aggressive nature in children as they are 3 times more likely to present with metastatic disease compared to adults [54]. However, due to the extreme rarity in this population, large scale studies to further examine these findings are lacking. This disease remains an individually case reportable event.

### 3.2. Familial Syndromes

Some children are at a greater risk of developing certain neuroendocrine neoplasms compared to the general pediatric population based on inherited genetic mutations. This includes but is not limited to multiple endocrine neoplasia (MEN) 1, MEN2A, MEN2B, familial medullary thyroid carcinoma (FMTC), Von-Hippel Lindau (vHL) syndrome, Neurofibromatosis type 1 (NF-1), and hereditary paraganglioma-pheochromocytoma syndrome. MEN1 results from a genetic mutation in the MEN1 gene which encodes the Menin protein [55]. Around 15% of patients with MEN1 are diagnosed in childhood and should be screened for the associated malignancies. Children with MEN1 most frequently present with pituitary (prolactin-secreting) and pancreatic (gastrinomas, insulinomas, and non-functional) NENs, in addition to primary hyperparathyroidism which is not currently classified as an NEN [56]. Children with MEN1 are not limited to NENs in these locations as there have also been documented cases of MEN1 patients with thymic, bronchopulmonary and adrenal NENs [56,57]. Given the heterogeneity in disease manifestation in this genetic syndrome, clinical management is complex and patient specific [58].

Akin to MEN1, MEN2A and 2B are also inherited in an autosomal dominant pattern, except they encompass the diseases that arise from a mutation in the RET gene [59]. There is essentially a complete penetrance of medullary thyroid cancer in MEN2A and MEN2B patients which may present as a thyroid nodule [59]. As such, children with MEN2A or 2B are evaluated for a medullary thyroid cancer risk profile and may undergo prophylactic thyroidectomy at or before 5 years old [60]. MEN2A and MEN2B patients are also at increased risk for pheochromocytomas and should begin screening at 11–16 years old depending on risk profile, determined by the specific RET mutation (Table 2) [2,60].

vHL results from a mutation in the VHL tumor suppressor gene which encodes the VHL protein, a regulator of angiogenesis [61]. Among many other malignancies, children with vHL are at an increased risk for pheochromocytomas, paragangliomas, and pancreatic NENs [15,62,63]. Children with NF-1 (mutation of the NF1 gene) are also at an increased risk for pheochromocytomas [43]. Pancreatic NENs are rare in NF-1 patients but have been reported in young adults with NF-1 [37,64].

### 3.3. Biochemical Work Up

Overall, NENs encompass a wide array of diseases and can manifest anytime in childhood and adulthood with a diverse symptom profile. Clinicians should be thoughtful in evaluating children with abnormally persistent symptoms, no matter how vague, as potential for an NEN. A clinical suspicion for an NEN warrants a biochemical diagnostic work-up and, if indicated, imaging. Many NENs are not diagnosed until histologically proven on a pathology sample. There are different markers that distinguish NENs from other types of tumors. Immunohistochemical staining for chromogranin-A (Cg-A) and synaptophysin are used for diagnosis of NEN, but some NENs express different granins than Cg-A potentiating misdiagnosis [3]. Cg-A can also be detected in plasma and serum but some authors have noted issue with low sensitivity (67–93%) and specificity (85%–96%) depending on the diagnostic method employed [65]. Similar issues with low sensitivity but higher specificity has been noted in using neuron-specific enolase for NEN diagnosis [66]. Considering the pitfalls with the currently utilized biochemical tests, there is need for better and more reliable biomarkers. At time of this publication there is an ongoing prospective multi-center trial investigating human circulating progastrin (hPG80) as a biomarker for the monitoring of NENs (NTC04750954). hPG80 has been demonstrated to be elevated in the plasma of patients with a variety of low- and high-grade NENs when compared to healthy controls [67]. However, its wide-spread use is pending validation.

There are a variety of different peptides that can be detected in the blood or urine of patients who present with functional tumors. Urinary 5-Hydroxyindolacetic Acid (5-HIAA) is a metabolite of serotonin which is helpful in the diagnosis of serotonin-producing tumors with higher sensitivity for midgut NENs compared to others [66]. 5-HIAA has high specificity, but serum serotonin and Cg-A can be used in conjunction for diagnosis [68]. Recently, a post-hoc analysis in the CLARINET study showed that 5-HIAA may have use in the diagnosis and monitoring of patients with nonfunctional NENs [69]. However, this has not been adopted into practice guidelines. For medullary thyroid cancer, serum calcitonin measurements are useful in disease detection as well as recurrence screening [60]. The functional pancreatic NENs can also be diagnosed with biochemical work up. Fasting gastrin levels and secretin stimulation tests can be used in the diagnosis of gastrinomas [70]. Classically, it has been advised to stop a proton-pump inhibitor prior to the stimulation test as it confounds the results. However, recent literature suggests this may be unnecessary [71]. Interestingly, the use of proton-pump inhibitors has also been shown to falsely elevate Cg-A levels, confounding that diagnostic test as well [72]. Over 95% of insulinomas can be diagnosed by serial plasma glucose and insulin levels during a 72 h fast showing hyperinsulinemic hypoglycemia [73]. Similarly, somatostatin can be measured to aid in the diagnosis of somatostatinoma and VIP can be measured in the plasma to diagnose a VIPoma in conjunction with hypokalemia and achlorhydria [74,75]. Functional pituitary NENs can be detected by elevated levels of growth hormone, ACTH, IGF1, or prolactin [31].

### 3.4. Imaging

There are several imaging modalities useful for the diagnosis of NENs. With the increased use of cross-sectional imaging, more and more NENs are incidentally found on computed tomography (CT) and magnetic resonance imaging (MRI) [76]. While this trend has been noted as the etiology of the increased incidence of NENs in the adult population, the same argument does not hold true in the pediatric population where frequent imaging is less likely to occur in accordance with limiting exposure to radiation (i.e., CT). However, when an NEN is suspected in a child or adolescent, undergoing the appropriate diagnostic imaging is important in establishing a timely diagnosis.

Imaging modalities, used alone or in combination with each other, include CT, MRI, positron emission tomography (PET), meta-iodobenzylguanidine (MIBG) scintigraphy, and somatostatin receptor (SSR) scintigraphy. For pituitary NENs, MRI is preferred to distinguish a pituitary lesion from surrounding soft-tissue lesions, whereas CT and the functional imaging studies are of less widespread use but may be appropriate for a select subset of patients with pituitary NENs [77]. In general, CT and MRI alone can help distinguish between well and poorly differentiated NENs based on enhancement, there remains other, more sensitive imaging modalities for the detection and diagnosis of NENs. MIBG is over 90% sensitive for the diagnosis of neuroblastomas, pheochromocytomas, and paragangliomas, but less sensitive in the detection of other NENs (e.g., pancreatic, midgut, non-catecholamine producing tumors) [78,79]. SSR scintigraphy uses a radiolabeled somatostatin analogue (111In-octreotide) to detect NENS by exploiting the elevated expression of somatostatin receptors on the majority of NEN tumor cells. However, SSR-PET/CT and SSR-PET/MRI have improved diagnostic quality compared to scintigraphy alone [80]. In pediatrics, SSR-PET is the imaging modality of choice given its decreased radiation dose and faster study time without compromising diagnostic sensitivity [81]. 68-Gallium-DOTATATE/DOTATOC/DOTANOC are the most commonly used SSR analogues used for diagnostic purposes today [79]. 64-Copper-DOTATATE is another radiolabeled SSR analogue FDA approved for use in the diagnosis of NENs which has been shown to be better at detecting NENs when compared to gallium [82]. In general, SSR-PET is limited in its ability to detect NENs that do not express high levels of somatostatin receptors as well as its ability to distinguish small NENs from surrounding tissue with elevated somatostatin receptors (i.e., pituitary NENs, inflammatory reactions) [77,83]. The benefit of functional imaging with MIBG or SSR-PET is the tumors detectable with these imaging modalities may benefit from therapeutic targeted radioisotope treatments (discussed further in the treatment section below). Additionally, some of these imaging modalities have the ability to distinguish NENs based on differentiation. Well-differentiated NENs demonstrate greater uptake with 68-Gallium-DOTA-peptide PET/CT, whereas poorly-differentiated NENs have low uptake. The reverse is observed with use of 18F-fluorodeoxyglucose (FDG)-PET/CT, an imaging modality which measures glucose metabolism. This phenomenon is well described in gastroenteropancreatic NENs, and has shown to be beneficial in assisting in classifying lesions in tandem with mitotic rate and Ki-67 [84]. In general, the diagnosis of a NEN in a pediatric patient typically involves a combination of these imaging modalities directed by clinical and biochemical work up [2].

### 3.5. Biopsy

Biopsy can be useful in the diagnosis and characterization of certain NENs. Biopsy can be used in bronchopulmonary NENs if anatomically accessible but may not be sufficient to distinguish between typical and atypical tumors [85,86]. When anatomically accessible, endoscopic ultrasound with fine needle aspiration can be used in gastric, proximal small bowel, and pancreatic NENs [87]. Biopsy does not have reliable sensitivity in pancreatic lesions less than 2 cm and also may mischaracterize neoplasms as Ki-67 indices can vary throughout the tumor itself [88,89]. Biopsy is helpful in the diagnosis of hepatic metastasis and may be necessary if the primary lesion is unknown [90]. Contrarily, biopsy of small bowel NENs are not typically feasible and are not performed but instead are diagnosed through the biopsy of hepatic metastases or surgical excision of the primary lesion [91]. Biopsy of suspected pheochromocytomas and paragangliomas are unique for NENs as it is actively not recommended to biopsy these lesions for diagnosis unless absolutely necessary. This is due to their increased risk of bleeding from hypervascularity, tumor seeding, and hypertensive crisis. If biopsy is necessary, it should only be done so once the patient is satisfactorily α-blocked to reduce the risk of hypertensive crisis [92].

As with any procedure, risk and benefit must be weighed prior to proceeding. If the biochemical and imaging work up is equivocal, biopsy can be beneficial in establishing a diagnosis. However, there are reports of inducing carcinoid crisis and even death from performing biopsies of NENs, but this is very rare [93,94,95]. Thus, biopsy may not be advisable if the diagnosis is already established and the results from which will not affect treatment planning. In the spirit of personalized medicine, the emergence of a “liquid biopsy” is gaining traction in the evaluation of malignancies. Liquid biopsies have the advantage over the typical biopsy techniques as they are less invasive, faster, and can collect more than sufficient volume of specimen for multiple analyses [96]. The NETest analyzes blood samples for circulating neuroendocrine genomic analytes (i.e., mRNA) and has been used in the evaluation of pheochromocytomas, paragangliomas, bronchopulmonary, gastrointestinal, and pancreatic NENs [97,98,99,100]. In the prospective study, NETest had a 99% accuracy in diagnosing NENs [100]. Some have also shown its benefit in predicting response to therapy and assessing for residual disease [101,102]. The role of this study in the pediatric population is uncertain.

## 4. How to Treat

### 4.1. Surgery

While complete surgical resection is the ideal treatment for most NENs, careful observation is appropriate for certain subsets of NENs. Infants less than 6 months old with small neuroblastomas can safely be observed as these tumors in this age group can spontaneously regress [103]. While not associated with regression, asymptomatic, non-functional pancreatic NENs less than 2 cm in size can also be observed [104,105]. Some have argued there is an unacceptable risk of disease progression and metastasis with observation [106]. As children have a longer amount of time to potentially develop progression of disease, it may be beneficial to excise these small tumors, but this remains controversial.

Technique for surgical excision is dependent on the location and grade/stage of the disease. Appendiceal NENs are often diagnosed on pathology after appendectomy for presumed appendicitis in children. With guidelines extrapolated from adult observations, it has classically been advised that appendiceal NENs greater than 2 cm should undergo right hemicolectomy due to risk of metastasis with special consideration for radical surgery in other tumors with high-risk features (e.g., high mitotic rate, high Ki-67 proliferation index, incomplete resection, lymph node involvement, and tumor at the base of the appendix) [107,108]. Yet, there are multiple studies in pediatric patients showing no survival advantage nor difference in disease progression for children undergoing simple appendectomy versus right hemicolectomy [109,110,111,112,113]. Thus, making the argument that radical surgery may not be beneficial in this population.

There is an emphasis on early detection for all NENs as complete surgical resection is curative in early locoregional disease [2,114]. This includes partial pneumonectomy (e.g., sleeve resections and bronchoplasty) for bronchopulmonary NENs and partial pancreatectomy (e.g., pancreaticoduodenectomy and distal pancreatectomy) for pancreatic NENs [115]. Even in local disease, surgery for an NEN can be complicated by severe reactions that can occur during surgical resection and anesthesia. Carcinoid crisis causes drastic and sudden hemodynamic instability from the sudden release of vasoactive peptides [116]. The risk of carcinoid crisis during procedures and surgical resection is not negligible as it has a reported incidence of 19%, with greater likelihood in patients with hepatic metastases [116]. While rapid administration of intravenous octreotide is the treatment for carcinoid crisis, recent studies have shown the prophylactic use of octreotide does not decrease the risk of developing perioperative/periprocedural carcinoid crisis [116,117,118]. Similar to carcinoid crisis, hypertensive crisis can occur in the perioperative/periprocedural time in patients with pheochromocytomas and paragangliomas due to sudden release of catecholamines [119]. Hypertensive crisis has been reported in children [120,121]. As such, children with catecholamine secreting tumors should undergo α-blockade with phenoxybenzamine or doxazosin followed by β-blockade prior to any procedures/operations [122]. Similar to adults, children should be sufficiently α-blocked prior to administration of β-blockers to prevent unopposed α-stimulation [123].

There are a variety of treatment options for metastatic NEN, but unfortunately none are curative like complete surgical resection. In patients with resectable primary lesion and metastatic disease (i.e., hepatic metastases), complete surgical resection of both with cytoreductive surgery is recommended [124]. In the setting of unresectable metastatic disease with resectable primary tumor, debulking by excision of the primary tumor is associated with increased 5-year survival in midgut NEN adult patients [125]. Hepatic metastases can also be managed with transarterial embolization (bland, radiation, or chemotherapeutic), selective internal radiation, or ablation [124,126]. Radiation and radiopharmaceuticals may be effective for primary and metastatic disease depending on the tumor characteristics detailed in the next section. Overall, given the diversity of disease processes encompassed under the classification of NEN, all pediatric patients with NENs should be discussed with a multi-disciplinary team with expertise in the management of pediatric NENs prior to surgical/procedural intervention.

### 4.2. Radiation Therapy

Radiation therapy and therapeutic radiopharmaceuticals are typically reserved for the treatment of unresectable or residual NENs, or patients not medically appropriate for surgical intervention [127,128]. External beam radiotherapy and/or stereotactic radiosurgery are used in aggressive, residual, and unresectable pituitary NENs, medullary thyroid carcinoma, middle ear NENs, thymic NENs, bronchopulmonary NENs, and gastroenteropancreatic NENs [60,129,130,131,132,133,134]. It is rare to use radiation therapy as a single therapeutic agent for any NEN but instead used in multimodal treatment plans.

Patients with norepinephrine and somatostatin receptor avid tumors detected with the functional imaging discussed above can be treated with similarly structured therapeutic radiopharmaceuticals. Nonresectable neuroblastomas, pheochromocytomas, and paragangliomas with elevated MIBG uptake are often treated with 131I-MIBG alone or in combination with chemotherapeutics [135,136,137]. A therapy still in evolution that offers improved outcomes over 131I-MIBG, is peptide receptor radionuclide therapy (PRRT). PRRT uses radiolabeled somatostatin analogues to deliver radiation therapy directly to the tumor. It is suitable for the treatment of NENs which overexpress somatostatin receptors [138]. Two different radiopeptides, 90Y-DOTATOC and 177Lu-DOTATATE, are in use today. PRRT has shown increased overall survival, progression free survival, event free survival, and response to treatment when compared to 131I-MIBG in the treatment of advanced pheochromocytomas and paragangliomas [139]. PRRT has also been used in MIBG refractory neuroblastoma [140,141,142]. The NETTER-1 trial showed that 177Lu-DOTATATE improved both progression free survival and quality of life [143]. At this time, 177Lu-DOTATATE is not FDA approved for use in children and is only used in clinical trials on a case-by-case basis. Prior to the use of either 131I-MIBG or PRRT, functional imaging proving the tumor’s avidity for these radiotherapeutics should be confirmed. Additionally, the NETest may be of use in predicting PRRT response and monitoring disease status during treatment [101,144].

As it stands, these therapies remain a palliative option for the treatment of advanced NENs. It is important to recognize these therapies do have potential adverse effects. Similar to operations and procedures, PRRT can induce carcinoid crisis and MIBG therapy can induce hypertensive crisis in NEN patients [145,146,147,148]. Although, these occurrences are documented in adults and it is unclear what the true risk is for pediatric patients.

### 4.3. Medical Management

Similarly, to how somatostatin analogues are beneficial in imaging and radiotherapy, they can also be used in treatment and symptom control. Octreotide and lanreotide are somatostatin analogues which have long been used to control carcinoid symptoms and reduce disease progression in patients with NENs [149,150,151]. The CLARINET trial demonstrated significant improvement in progression free-survival of 65.1% at 24 months in NEN patients treated with lanreotide versus only 33.0% in the placebo group [151]. The PROMID trial found octreotide was beneficial in prolonging time to tumor progression [152,153]. There are multiple studies evaluating these agents and others for disease control and symptom management, including the TELESTAR study which found telotristat ethyl improved carcinoid syndrome diarrhea and reduced urinary 5-HIAA levels in patients not well controlled with somatostatin analogues [154]. However, response is dependent on somatostatin receptor status of the tumor and PRRT has been shown to be more effective than somatostatin analogues alone in the treatment of adult NEN patients [143].

Many of the treatment options available to pediatric NEN patients is extrapolated from what works in adults [155]. Various chemotherapeutics (cyclophosphamide, vincristine, dacarbazine, temozolomide, capecitabine, etoposide, cisplatin/carboplatin, everolimus, and mTOR inhibitors) have been used with varying success to treat advanced NENs [60,156,157,158,159,160,161,162]. The use of chemotherapeutics in pediatric patients must be decided on an individual basis with consideration of inclusion in clinical trials suited to the location and molecular profile of the tumor.

### 4.4. Clinical Trials

Due to the rarity of NENs in the pediatric population and the wide variety of organ systems they can originate from, there are few clinical trials that have been conducted or are being conducted specifically in this population beyond those focused on radiopharmaceuticals. In general, pediatric patients with NENs have and continue to be treated in solid tumor clinical trials that involve both pediatric and adult populations typically geared toward other solid malignancies. In a form of concordance with the movement towards personalized medicine, many of these patients are included in clinical trials if they have refractory or relapsed disease with certain genetic mutation profiles that can be targeted with the investigational chemotherapeutics (e.g., MDM2, MDMX, RET, BCL-2, and many others) [163]. Current ongoing therapeutic clinical trials focused on NENs are investigating the clinical efficacy and safety of 177Lu-DOTATATE in children with gastroenteropancreatic NENs, paragangliomas, and neuroblastomas (NCT04711135, NCT03966651) [164,165,166]. Prior Phase I and II clinical trials with 90Y-DOTATOC and 177Lu-DOTATATE in children with solid tumors have shown minimal dose-limiting toxicities with a good safety profile indicating their safety in this population [167,168]. As these ongoing clinical trial progress to completion, the treatment paradigm of pediatric NENs are expected to change accordingly. In the interim, a pediatric patient with refractory, recurrent, and/or retained neuroendocrine disease should be considered for inclusion in a clinical trial.

## 5. Long-Term Outcomes

Overall, pediatric cancer patients are living longer with more cancer survivors living well into adulthood [169]. Although many childhood cancers survival rates have seen large improvements over the past few decades, only a very modest improvement in survival has been seen in some pediatric NEN patients. For instance, from 1975 to 2006, 5-year survival for neuroblastomas increased nearly 30 percent whereas over the same period, 5-year survival only increased by 1 percent for other pediatric NEN patients. Overall, pediatric NEN patients do have a better survival compared to adults, but survival varies widely based on site of the primary lesion [6]. Survival for appendiceal NEN is observed to be 100% whereas foregut NEN survival is only 26%, and even worse for those of unknown primary (10.5% observed survival rate) [12]. Given this significant heterogeneity in survival outcomes and the individual rarity for each tumor location, it is not surprising that clinical trials and studies do not capture pediatric NENs as an individual entity like they do for other pediatric malignancies (i.e., leukemias, lymphomas, etc.).

Similar to the wide margin of survival for all pediatric NENs, rate of recurrence is equally tumor and organ specific. Of the multitude of studies on pediatric appendiceal NENs, most report no observed recurrences in their patient populations [109]. One study out of Poland has documented a recurrence after surgical resection (which was subsequently surgically removed, with no further evidence of recurrence on follow-up) [170]. In contrast, other extra-appendiceal gastrointestinal NETs, although rarer than appendiceal NETs in children, have a greater risk of recurrence [13]. Within the adult and pediatric population, recurrence of middle ear NEN is quoted to be 22% [171]. There are multiple small case series and retrospective reviews on bronchial NENs in mixed pediatric and adult populations where no recurrences were detected and others finding a recurrence rate of 2–27% depending on histological subtype (atypical 7.9 times more likely to recur than typical) [13,22,172]. The recurrence rate for medullary thyroid cancer even after total thyroidectomy is 9–12% in children but varies greatly in timing of thyroidectomy and genetic predisposition [60,173]. Children with MEN 2 mutations who undergo prophylactic thyroidectomy at younger ages have lower risk of recurrent or persistent disease [173]. Hence the recommendation for prophylactic thyroidectomy at or before 5 years old in children with high risk MEN2 mutations [60]. Recurrence of NENs can occur even 50 years after initial diagnosis making life-long surveillance a necessity for childhood NEN survivors [174].

The North American and British Childhood Cancer Survivor Studies (CCSS and BCCSS) have been instrumental in defining the long-term outcomes for a number of pediatric malignancies, but unfortunately NENs have not been characterized through this study [175,176]. Thus, the long-term outcomes of survivors of pediatric NENs as they live into adulthood are relatively unknown. Through the childhood cancer survivor studies, it was found that childhood cancer survivors develop more chronic health conditions (cardiac, musculoskeletal, neurologic, endocrine, and gastrointestinal) and are at higher risk of developing a subsequent malignancy compared to the general population [169,177]. Childhood cancer survivors also face social and economic disadvantages in life as they are less likely to graduate college and less likely to have full-time employment [178,179]. Almost 15% of childhood cancer survivors develop posttraumatic stress symptoms that can impede quality of life [180]. Many of these risks differ based on the primary malignancy, however the treatment modalities employed during childhood also confer risk of developing these outcomes in adulthood. Although these studies do not pertain to pediatric NEN patients in particular, it is unlikely pediatric NEN patients are exempt from these trends as the treatment modalities are similar to other childhood malignancies. These patients should be monitored appropriately for outcomes of the like well into adulthood.

## 6. Challenges and Opportunities

Like most rare diseases, relatively low incidence has been a significant deterrent to advancements in clinical drug development in NENs. Many clinical trials are often terminated early due to lack of enrollment. NEN translational research also suffers from lack of easily accessible high quality pre-clinical models. Last but not the least, dichotomization of NENs among various site-specific disease groups has led to a lack of common terminology, classification, and management framework. While we acknowledge the above mentioned challenges, we are also optimistic about the road ahead. The several fold increase in the incidence of NENs has garnered attention of not only the pharmaceutical industry but also the National Cancer Institute (NCI) and has resulted in a significant upsurge in interventional therapeutic clinical trials [181]. Consensus is being generated to homogenize terminology and classification of various subsets of NENs. Our understanding of the molecular underpinning of NENs has substantially improved in the last decade and molecular characterization of NENs is not only being considered to classify the morphologic subtypes of NENs but will also pave the way for relevant precision medicine clinical trials in the future.

## 7. Conclusions

NENs can present in a variety of ways with outcomes which range from benign to very aggressive. It is crucial healthcare providers of all levels in all specialties be aware of how NENs can present given the potential for high morbidity in delayed diagnoses. As our understanding of this disease continues to progress, the management of NENs continues to evolve. This review provides an overview of all the commonly encountered forms of NENs in pediatric patients with up to date recommendations so any healthcare provider can quickly and accurately acclimate themselves.

## Figures and Tables

**Figure 1 cancers-14-05049-f001:**
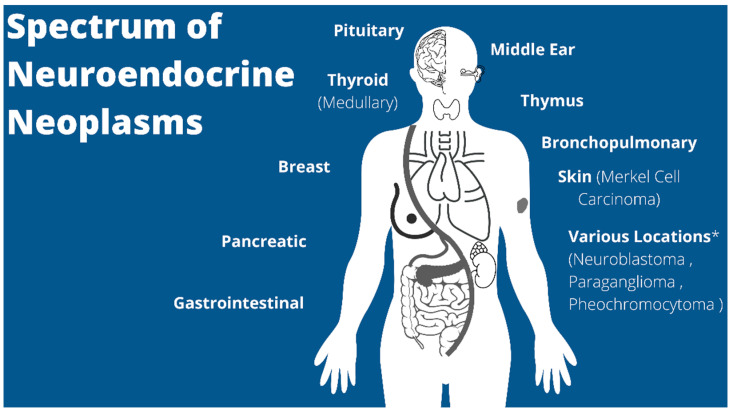
Illustration of some of the common and less common neuroendocrine neoplasms encountered in the pediatric and adult populations (* most commonly originate from the adrenal glands; paragangliomas are extra-adrenal).

**Table 1 cancers-14-05049-t001:** Classification of neuroendocrine neoplasms adapted from the WHO classifications up through 2022 as described previously [5,6].

**Neuroendocrine Neoplasms**	**Grade**	Terminology/Differentiation/ Location	Mitotic Rate (per 2 mm^2^)	Ki-67 (%)
	**Low**	Grade 1, well-differentiated NET (extra-thoracic)	<2	<3
		Typical Carcinoid (Thoracic)	<2	–
	**Intermediate**	Grade 2, well-differentiated NET (extra-thoracic)	2–20	3–20
		Atypical Carcinoid (Thoracic)	2–10	–
	**High**	Grade 2, well-differentiated NET	>20	>20
		Poorly differentiated Carcinoma (small cell and large cell)	>20 (thoracic > 10)	>20

**Table 2 cancers-14-05049-t002:** Screening recommendations in children with known familial syndromes at increased risk for developing different NENs. Modified from the 2015 American Thyroid Association guidelines and the 2021 NCCN Neuroendocrine and Adrenal Tumors guidelines [2,60]. (CT = computed tomography, MRI = magnetic resonance imaging, EUS = endoscopic ultrasound, CEA = carcinoembryonic antigen).

**Familial Syndromes**	**Syndrome**	**NEN**	**NEN** **Percent Occurrence**	**Screening Recommendations**	
	**MEN 1**	**Pancreatic** **(gastrinoma,** **insulinoma)**	**20–80%**	– chromogranin-A, pancreatic polypeptide, glucagon, VIP annually starting at 8 years old – fasting gastrin annually starting at 20 years old – consider abdominal CT, MRI, or EUS every 3–5 years starting at 20 years old	
		**Pituitary** **Adenoma** **(prolactinoma)**	**30–40%**	– serum prolactin, IGF-1, fasting glucose and insulin annually starting at 5 years old – head MRI every 3–5 years starting at 5 years old	
	**MEN 2A/2B**	**Medullary Thyroid** **Cancer**	**≥98%**	Highest Risk	Prophylactic thyroidectomy at or before 1 year old with physical exam, neck ultrasound, serum calcitonin/CEA every 6 months for 1 year and annually thereafter (*serum calcitonin confounding in infants as normally elevated)
				High Risk	Physical exam, neck ultrasound, serum calcitonin annually starting at 3 years old; Prophylactic thyroidectomy at or before 5 years old based on serum calcitonin followed by physical exam, neck ultrasound, serum calcitonin, and CEA every 6 months for 1 year and annually thereafter
				Moderate Risk	Physical exam, serum calcitonin every 6 months for 1 year and annually thereafter if calcitonin remains normal; prophylactic thyroidectomy when calcitonin levels elevated
		**Pheochromocytoma**	**≥50%**	Highest/ High Risk	Free plasma metanephrines/normetanephrines or 24 h urine fractionated metanephrines annually starting at 11 years old. Adrenal imaging with CT/MRI if elevated
				Moderate Risk	Free plasma metanephrines/normetanephrines or 24 h urine fractionated metanephrines annually starting at 16 years old. Adrenal imaging with CT/MRI if elevated
	**Von-Hippel Lindau**	**Pheochromocytoma**	**10–20%**	– Blood pressure at all medical visits starting at 2 years old – Free plasma metanephrines/normetanephrines or 24 h urine fractionated metanephrines annually starting at 5 years old – abdominal MRI or CT with and without IV contrast every 2 years starting at 15 years old	
		**Paraganglioma**	**10–20%**		
		**Pancreatic**	**5–17%**		

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
