# Peer review of "Pediatric Neuroendocrine Neoplasms: Rare Malignancies with Incredible Variability"

_cancers, 2022, doi:10.3390/cancers14205049_

Round 1
Reviewer 1 Report
In this review, authors provided an overview of variable aspects of pediatric neuroendocrine neoplasms from classification, diagnosis to treatment and outcomes, which is informative and self-explanatory.
I have some specific comments for the authors to address.
1. While the gastroenteropancreatic NENs were classified according to only the mitotic rate and Ki-67 index (grade 3 (G3) tumors incorporated both well-differentiated (WD) tumors and poorly differentiated (PD) tumors), the revised 2019 WHO system classified gastroenteroapcnreatic NEN as G1, G2, and G3 NET (previously NEC with WD) and NEC, and the treatment strategies for NEC and WD G1/2/3 NET were also explicitly separated. The 8th edition of the American Joint Committee on Cancer (AJCC) staging manual separated the TNM staging system for NEC from WD G1/2/3 NET of the pancreas in that NEC follows staging system of exocrine tumors of the pancreas while WD G1/2/3 NET follows staging system of neuroendocrine tumors of the pancreas. Although this review was focused on pediatric patients in which the pancreatic NEN is less common than the adult, this is a recent substantial change and should be described in this review.
2. It is true that the current classification of NENs differes between organ systems and casuses considerable confusion. To address this, a common framework for NEN classification was proposed by the International Agency for Research on Cancer (IARC) and WHO expert consensus meeting which suggested distinction between differentiated NETs and NECs, irrespective of their site of origin. Authors could introduce this proposal in their review.
3. Page 4, line 120
Itt is confusing whether the most common sites of the ectopic ACTH-producing NENs are bronchus or pancreatic, or pituitary gland. Please further clarify this.
4. Page 7, "Imaging" section
4-1) The distinction between NETs and NEC based on imaging features is clinically relevant in view of the difference in patient management, and should be included here. First, in the cross-sectional imaging (i.e., CT and MRI), the enhancement degree differs between NET (well-enhanced) and NEC (less-enhanced and difficult to differentiate from other poorly differentiated tumor). Second, the use of PET/CT in NEN differes according to the tumor differentiation. 68Ga‐DOTA‐peptides PET/CT is the gold-standard functional imaging modality to study well-differentiated NENs in Europe and is included in European guidelines. In contrast, in case of high‐grade tumors, 68Ga‐DOTA‐peptides PET/CT has limited value and FDG-PET/CT may be useful.
4-2) The use of endoscopic ultrasound for gastric, proximal small bowel, and pancreatic NENs should be added.
5. Page 8~10, "How to treat"
The treatment of NEN significnatly differs according to the histologic grade and staging of the tumor. The management strategies, including both chemotherapy and radiation therapy, for well-differentiated NETs are completely different from those for poorly differentiated NEC. I recommend authors to further elaborate this part by providing separate discussion of treatment, according to the resectability (i.e., localized resectable disease, resectable primary and metastatic disease, and unresectable disease) and tumor differentiation (i.e., NET and NEC). A flowchart might be useful.
Reviewer 2 Report
Dear Author,
This is an interesting paper.
Here are my observations/questions/comments:
1. Title. I suggest not using “review” because the paper will be “Review”/ “A review of….” Thus the type of article is clear from the start
2. …“most up to date” is a pleonasm
3. Table 1 – The classifications for 2015-2021 are not exactly the same, neither the terminology
4. Usually, red Font is not recommended (Table 1, Figure 1)
5. Figure 1 = You introduced 8-9 sites and 4 types of tumors. I suggest you use “skin” and mark with an index the type of tumor and, also, medulla of the adrenals or enterochromafin cells (and mark with an index the types of the tumors)
6. Lines 116-118 – NEN cannot be associated with Cushing’ s syndrome of exogenous sources since this is a iatrogenic type. Cushing syndrome, as carcinoid syndrome, may be paraneoplasic due to non-pituitary (tumor-related) ACTH over production.
7. Paraneoplasic (NEN- associated) Cushing syndrome is not typical with weight gain as opposite to other types of Cushing syndrome (non-NEN). Also, a particular clue for pediatric Cushing syndrome involves growth issues – line 119
8. Lines 125-126 are not clear. There is a big difference between Cushing disease (underling a PITNET with pituitary ACTH excess) and paraneoplasic Cushing syndrome (underling a NEN of any location, but with non-pituitary, ectopic ACTH production)
9. Neuroblastoma – you should mention from the start the aggressive profile and the mix forms neuroblastoma-ganglioneuroma
10. Chapter 3.2. In endocrinology, an important aspect involves medullary thyroid carcinoma (with strong genetic background), thus this type of NEN might display a presentation as a thyroid nodule
11. Table 2. With concern of RET mutations underling medullary thyroid carcinoma and other components of MEN2A, the main clue of classification is the RET mutation, not the clinical or paraclinical aspects – there are specific guidelines that indicate the age of prophylactic thyroidectomy depending on RET exons
12. Chapter 3.c. There are numerous hormonal assays – I suggest you mention those related with the prior chapters – like calcitonin, neuron specific enolase, serotonin in carcinoid syndrome, etc.
13. Chapter 4.b. Do you mean radionuclide therapy (which is different for radiotherapy)?
14. Chapter 4.c. You only mentioned CLARINET for lanreotide, but there are more studies actually on octreotide (CLARINET had some limits concerning the studied population)
15. Discussion. I suggest you introduce some ideas/concepts concerning current limits of the domain and further expansion.
16. Also, I suggest a section at the end of Introduction to specify the aim of the paper, and a separate section to specify the Methods including the type of study, cites papers, etc.
17. Conclusion – This should be a useful take home message
Thank you very much,
Best regards,
Reviewer 3 Report
This is a comprehensive review of neuroendocrine neoplasms in children and adolescents including multiple aspects across various organ sytems intended to raise awareness and provide guidance for healthcare providers. Given the rarity of neuroendocrine neoplasms, the ubiquitous sites of origin and different specialists involved, the review constitutes a valuable summary of presenting signs and symptoms, hereditary syndroms, laboratory work-up, imaging and treatment options.
Comments:
Simple summary and abstract.
The authors state in an alarming way without offering an explanation that NENs have increased 7 fold in the past few decades. Only later in the text they suggest that the increase might be attributable to improved diagnostic tools and imaging and that epidemiologic data for children are missing. I would attenuate this statement since this is not the main reason why healthcare providers should consider NENs in their diagnotic work-up
Line 78. I would write Diagnosis rather than Making the Diagnosis
Line 228: delete „use of“ or write „has“
Line 268 a NEN
Line 274 and 274: This sentence sounds incomplete or too colloquial
Line 398 and 399. Please include etoposide and cisplatin or carboplatin to the list of chemotherapeutics in use and add the respective literature.
Please update the list of references that should also include literature published in 2022 such as Rindi G et al. WHO Classification Endocr Patholo 2022 and Caroline Pietermann et al. MEN 1 guidelines Clin Endocrino 2022
